# Programmed Death-Ligand 1 (PD-L1) as Immunotherapy Biomarker in Breast Cancer

**DOI:** 10.3390/cancers14020307

**Published:** 2022-01-08

**Authors:** Martín Núñez Abad, Silvia Calabuig-Fariñas, Miriam Lobo de Mena, Susana Torres-Martínez, Clara García González, José Ángel García García, Vega Iranzo González-Cruz, Carlos Camps Herrero

**Affiliations:** 1Department of Medical Oncology, Hospital General Universitario de Valencia, 46014 Valencia, Spain; m.lobodemena@gmail.com (M.L.d.M.); claragargon@gmail.com (C.G.G.); camps_car@gva.es (C.C.H.); 2Molecular Oncology Laboratory, Fundación Investigación, Hospital General Universitario de Valencia, 46014 Valencia, Spain; calabuix_sil@gva.es (S.C.-F.); susana.torres@alu.umh.es (S.T.-M.); 3Unidad Mixta TRIAL, Centro Investigación Príncipe Felipe-Fundación Investigación, Hospital General Universitario de Valencia, 46014 Valencia, Spain; 4Centro de Investigación Biomédica en Red Cáncer, CIBERONC, 28029 Madrid, Spain; 5Department of Pathology, Universitat de València, 46010 Valencia, Spain; 6Department of Pathology, Hospital General Universitario de Valencia, 46014 Valencia, Spain; joseangel.garcia.bis@gmail.com; 7Department of Medicine, Universitat de València, 46010 Valencia, Spain

**Keywords:** PD-L1, breast cancer, biomarker, prognostic value, immunotherapy, chemotherapy

## Abstract

**Simple Summary:**

Breast cancer (BC) is the most common malignant neoplasm in women and one of the leading causes of cancer death in women worldwide. Programmed death-ligand 1 (PD-L1) is becoming an emerging biomarker in BC in recent years. It has been correlated with worse outcomes in patients with hormone receptor positive, but it has a predictive role to guide response to systemic treatment in the triple-negative breast cancer (TNBC) subtype, especially in the metastatic setting. Immune checkpoint inhibitors are beginning to be a part of the treatment for many TNBC patients. However, more studies are needed in order to identify wherefore immunotherapy benefits TNBC patients regardless of PD-L1 status in the localized disease, but only offer an improvement for PD-L1 positivity expression in the advanced setting. The aim of this review is to analyze PD-L1 in all BC subtypes, including clinical trials with anti-PD-1/L1 and their results.

**Abstract:**

Breast cancer constitutes the most common malignant neoplasm in women around the world. Approximately 12% of patients are diagnosed with metastatic stage, and between 5 and 30% of early or locally advanced BC patients will relapse, making it an incurable disease. PD-L1 ligation is an immune inhibitory molecule of the activation of T cells, playing a relevant role in numerous types of malignant tumors, including BC. The objective of the present review is to analyze the role of PD-L1 as a biomarker in the different BC subtypes, adding clinical trials with immune checkpoint inhibitors and their applicable results. Diverse trials using immunotherapy with anti-PD-1/PD-L1 in BC, as well as prospective or retrospective cohort studies about PD-L1 in BC, were included. Despite divergent results in the reviewed studies, PD-L1 seems to be correlated with worse prognosis in the hormone receptor positive subtype. Immune checkpoints inhibitors targeting the PD-1/PD-L1 axis have achieved great response rates in TNBC patients, especially in combination with chemotherapy, making immunotherapy a new treatment option in this scenario. However, the utility of PD-L1 as a predictive biomarker in the rest of BC subtypes remains unclear. In addition, predictive differences have been found in response to immunotherapy depending on the stage of the tumor disease. Therefore, a better understanding of tumor microenvironment, as well as identifying new potential biomarkers or combined index scores, is necessary in order to make a better selection of the subgroups of BC patients who will derive benefit from immune checkpoint inhibitors.

## 1. Introduction

### Rationale

Breast cancer (BC) constitutes the most common malignant neoplasm in women and represents one of the leading causes of cancer lethality in this population [1].

A number of well-known biomarkers are employed in the management of BC, such as estrogen/progesterone receptor positivity and overexpression/amplification of HER2 (human epidermal growth factor receptor 2) [2]. However, the utility of programmed death-ligand 1 (PD-L1) as a predictive biomarker in all BC subtypes remains unclear. The PD-L1 immunohistochemistry (IHC) biomarker landscape is complex due to the heterogeneity of IHC assays, with diverse scoring algorithms approved for different scenarios and tumor indications [3]. Currently, the most commonly used anti-PD-L1 antibody clones are 22C3, SP142, SP263, 28-8, E1L3N and 73-10 [4,5,6]. Insufficient studies have evaluated the PD-L1 expression in tumor and immune cells in BC, and preliminary data are divergent with still great variability in the analysis. Moreover, 40–50% differences can be observed depending on the antibody used for its detection. It could be a certain degree of concordance between IC and CPS for the evaluation of PD-L1 across different assays; nonetheless, the four PD-L1 assays are analytically discordant [4]. PD-L1 expression by IHC in BC is low (around 10–30%) compared with other tumors, such as lung cancer, and varies with stage and the molecular subtype, being the highest expression in triple-negative breast cancer (TNBC), ranging from 30 to 60% [7,8].

In addition, predictive differences have been found in response to immunotherapy depending on the stage of the tumor disease.

## 2. Objectives

In this paper, we present a summary of the scientific literature and a bibliographic review of PD-L1 as a potential biomarker in BC. Studies evaluating PD-L1 status in all BC patients, as well as clinical trials with PD-1/PD-L1 checkpoint inhibitors, are included in this analysis. Finally, based on the previous literature, we propose a possible protocol for PD-L1 analysis in BC.

## 3. Methodology

We performed a review of the current literature about the utility of PD-L1 expression as a predictive or prognostic marker in BC. 

### Search Strategy and Study Selection Criteria

Clinical trials using immunotherapy with anti-PD-1/PD-L1 in BC, as well as prospective or retrospective cohort studies about PD-L1 in BC, were included. There were no limits on the language of publication, BC stage and subtype, or treatment regimen received. 

The search was conducted on different databases, including MEDLINE, EMBASE and Cochrane Library. It was updated to September 2021, with no restrictions on the date of publication. 

## 4. Results and Discussion

### 4.1. PD-L1 Pathway and PD-L1 Testing in General

PD-L1 or B7 homolog 1 (B7-H1) is a protein encoded by the CD274 gene in humans. PD-L1 is a type 1 transmembrane protein ligand with a significant immunoregulatory role by suppressing the immune system in physiological processes, such as pregnancy, antigen presentation to T lymphocytes, and tissue and organ transplants, as well as in pathological processes such as infectious and immune diseases, and especially in cancer. In general, PD-L1 expression is observed on B and T lymphocytes and antigen-presenting cells, as well as in some non-lymphoid tissues [9].

The immune system reacts under normal conditions to foreign antigens associated with exogenous or endogenous danger signals. This causes the proliferation of specific CD8 and CD4 T lymphocytes against these antigens. The binding of the PD-L1 ligand with its PD-1 or B7.1 (CD80) receptors transmits a suppressive signal to T lymphocytes that leads to a reduction in the proliferation and a decrease in the immune response [10].

PD-L1 transmits intracellular signals in the cells that promote cell proliferation and survival and protect against pro-apoptotic stimuli mediated by interferons [11,12]. Interferon gamma (IFNγ) produced by T cells causes the activation of the Janus kinase (JAK) signal transducer and activator of transcription (STAT) pathway. This results in transcriptional activation of interferon regulatory factor 1 (IRF1), which finally binds to the PD-L1 promoter [13]. Tumor necrosis factor alpha (TNFα) and also IFNγ activate the NF-κB pathway that can transcriptionally transactivate PD-L1 transcription. These pathways supply an explanation for the high expression of PD-L1 in inflamed tissues, which includes highly infiltrated tumors. Nonetheless, the regulation of PD-L1 transcription is different depending on the cell type or physiological and pathological situation. Numerous mechanisms controlling the expression of PD-L1 remark on its differing roles, depending on the cell type or the location [13].

PD-1 is principally expressed in different cells of the immune system, and it has two ligands, namely PD-L1 (also called B7-H1 or CD274) and PD-L2 (B7-DC or CD273). Although the interaction of PD-1/PD-L2 shows a 2- to 6-fold higher affinity in comparison to the PD-1/PD-L1 interaction, PD-L1 is considered the major PD-1 ligand (Figure 1). The function of PD-L2 is less known, being principally an inhibitory molecule expressed not only by antigen-presenting cells, but also by other immune cells in an inducible manner, mainly through Th2-associated cytokines [14]. Its clinical role is the scope of the current investigation [15,16,17].

In diagnostic routine, PD-L1 expression is measured by using IHC [18]. PD-L1 IHC slide is evaluated semi-quantitatively by the pathologist. A neoplastic cell is counted as PD-L1-positive if there is a membranous staining, irrespective of staining intensity and whether the membrane depicts complete or partial PDL1 expression. If there is cytoplasmic but no membranous staining, a tumor cell is considered PD-L1-negative. However, for immune cells, either granular cytoplasmic or membranous staining is enough for a positive count [18,19,20]. The antibody clone used for IHC, the percentage of PD-L1-positive immune cells, and/or tumor cells and the scores should be reported by the pathologist.

The most relevant PD-L1 scores used in malignant tumors are tumor cell (TC), Tumor-Proportion Score (TPS), immune cell (IC), Immune-Cells Present (ICP), and Combined Positive Score (CPS) (Table 1) [7,18,21,22].

The tumor-cell score is defined as the percentage of the area covered by PD-L1-positive tumor cells in relation to the whole tumor area [18,23]. The Tumor-Proportion Score is another similar method to analyze PD-L1 that considers the percentage of viable tumor cells, showing PD-L1 partial or complete membrane relative to all viable tumor cells [18,24]. The immune-cell score is calculated as the proportion of tumor area, including associated intratumoral and contiguous peritumoral stroma, compounded by PD-L1 staining immune cells (T and B lymphocytes, macrophages, dendritic cells and granulocytes) of any intensity [18,25]. Additionally, Immune-Cells Present is used to determine positive immune cells, i.e., the percent area of ICP exhibiting PD-L1-positive immune-cell staining (also evaluated at any intensity) [25]. Finally, the Combined Positive Score is calculated based on the number of PD-L1-positive tumor cells and intratumoral immune cells or those in a narrow rim around the tumor (lymphocytes and macrophages; neutrophil granulocytes do not count), respecting total of viable tumor cells. The number of PD-L1-positive tumor cells and PD-L1-positive immune cells is summarized relative to the number of all viable tumor cells and then multiplied by 100. CPS is stated without any units. The maximum CPS is defined as 100.

The present PD-L1 IHC biomarker landscape is complex. Various IHC assays with heterogeneous scoring algorithms are approved for different therapies and tumor indications [3]. There is no unique standardized method for PD-L1 evaluation, since several scoring methods have been validated in clinical studies [18]. The clinical utility of PD-L1 testing have great variations between cancer types and treatment settings [22].

Currently, there are diverse developed PD-L1 assays to evaluate PD-L1 expression: SP142, SP263, 28-82, 22C3, E1L3N, 73-10, E1J2J, 5H1, 4059 and 9A11, among others [6,22,26,27,28]. The IHC assays that are most frequently used to determine the expression level of PD-L1 are PD-L1 IHC 22C3pharmDx Assay (PD-L1 22C3, DAKO) and SP142 Assay (PD-L1 SP142, VENTANA) (Table 2) [3,5,6,28,29]. Normally, the approval of immunotherapy drugs is often linked to a certain PD-L1 IHC assay. PD-L1 IC score (assay: SP142) was used for the approval of atezolizumab, whereas PD-L1 CPS (assessed with assay 22C3) is predictive for pembrolizumab. In most cases, PD-L1-positive expression is considered when TPS is 1% or higher, or CPS is ≥ 1 [3].

Various harmonization studies have tried to correlate the concordance between diverse PD-L1 IHC assays in different tumors, including BC, showing divergent data. The Blueprint project published in 2017 [32] collected a total of 39 non-small-cell lung cancer tumors. These samples were stained with four PD-L1 IHC assays (22C3, 28-8, SP142 and SP263). Three experts independently evaluated the percentages of TC and IC in the different assays. The results revealed that three of the four assays were closely aligned on TC staining (agrees in >85% of cases), whereas SP142 showed consistently fewer TCs stained (about 64% of concordance). All of the antibody clones demonstrated IC staining, but with greater variability than with TC staining [32]. The Blueprint2 project published in 2018 [33] was conducted by employing 81 different histological and sample types of lung cancer. The study compared diverse PD-L1 assays (22C3, 28-8, SP142, SP263 and 73-10). The slides were evaluated by an international panel of pathologists. The results showed highly comparable staining by the 22C3, 28-8 and SP263 assays; less sensitivity with the SP142 assay; and higher sensitivity with the 73-10 assay to detect PD-L1 expression on TC. However, there was a poor reliability in IC PD-L1 scoring between different clones [33,34,35].

In the TNBC field, SP263, 22C3 and SP142 assays were compared in a post hoc analysis of the IMpassion130 trial [36] that was published in 2021 by Rugo et al. [37]. To measure the PD-L1-positive status of the 614 TNBC samples, the standard cutoffs were IC ≥ 1% for SP142 and SP263, and CPS ≥ 1 for 22C3. The PD-L1-positive populations identified by 22C3 (81%) and SP263 (75%) were larger than SP142 assay (46%), showing that 22C3 and SP263 were not concordant with SP142. Moreover, investigators evaluated if using IC ≥ 1% as the cutoff with 22C3, instead of the standard CPS ≥ 1 used with this assay, would lead to analytical concordance with SP142, but they did not find greater concordance. Further, a mathematical model identified the optimal cutoffs for PD-L1-positive as CPS ≥ 10 for 22C3 and IC ≥ 4% for SP263. The concordance for harmonized cutoffs for SP263 (IC ≥ 4%) and 22C3 (CPS ≥ 10) to SP142 (IC ≥ 1%) was deficient (approximately 75%) [37].

In general, few studies show that it could be a certain degree of concordance between PD-L1 evaluation across some assays [32,33,34], but nowadays, they cannot be used interchangeably in clinical practice [21,29,30]. 

In general, the positivity of PD-L1 (using different assays and scores) has been associated with worse clinical outcomes and poor prognosis in many solid tumors, such as pancreatic [38], colorectal [39], gastric [40] or advanced lung cancer [41]. In some cases, such as early non-small-cell lung cancer, PD-L1 expression could involve better outcomes [42]. Clinical trials have shown that immunotherapy based on monoclonal antibodies targeting PD-L1/PD-1 prevent the inhibitory effects of the PD-1/PD-L1 pathway, thus improving T-cell functions. This can result in better survival, especially in patients with melanoma and non-small-cell lung cancer [13]. Furthermore, other tumors, such as bladder and renal cell carcinoma, may benefit from immunotherapy treatment. Nonetheless, the predictive value of PD-L1 expression in response to PD-1/PD-L1 inhibitors in some tumors, such as melanoma, renal, hepatocellular or small-cell lung cancer, is still unclear [12,43,44,45].

Apart from the PD-L1 IHC diagnostic assay, the detection of mRNA by in situ hybridization (ISH) as an alternative to detection of proteins is under investigation. A study published in 2019 by Duncan et al. included 90 patient samples with non-small-cell lung cancer, head and neck squamous cell carcinoma and urothelial carcinoma. The expression of mRNA was assessed by ISH, using the RNAScope 2.5 assay and probe CD274/PD-L1. The percentage of tumor cells with PD-L1 mRNA expression staining, using RNAScope, demonstrated statistical significance (*p*-value < 0.05) in the PD-L1 high (TPS ≥ 25%) vs. the PD-L1 low (TPS < 25%) groups for all tumors [46].

This review summarizes the current evidence of PD-L1 expression as a biomarker in BC and its possible association with clinical outcomes. We also discuss the potential value of PD-L1 in predicting the efficacy of different treatment options, including chemotherapy and immunotherapy, in some subtypes of BC, such as triple-negative tumors.

### 4.2. PD-L1 in Breast Cancer

BC is considered a heterogeneous disease with different molecular subtypes. The routine setting includes estrogen receptor (ER), progesterone receptor (PR) and epidermal growth factor receptor 2 (HER2) expression [47], which are used to classify BC into four different subtypes: luminal A (ER-positive, high expression PR, HER2-negative, Ki-67 less than 14%, low-risk gene signature); luminal B (ER-positive, HER2 negative, Ki67 ≥ 15%, low expression PR, high-risk gene signature); HER2 (HER2 positive) and triple negative breast cancer (absence of ER and PR expression and HER2 negative) [48]. These tumor types are associated with different treatments and clinical outcomes [49].

PD-L1 is usually not identified in normal breast tissue, even though its expression has been described in BC. PD-L1 expression in BC is measured in both tumor and immune cells. Few studies have evaluated tumor and immune cells’ expression in BC, but preliminary data are conflicting and there is still great variability in the analysis. Up to 40–50% differences can be observed depending on the antibody used for its detection [49]. Nonetheless, PD-L1 assays are not analytically concordant [4,50].

PD-L1 expression by IHC in BC is low (10–30%) compared with other tumors, such as non-small-cell lung cancer (around 70%) [51], and varies with stage and molecular subtype, with the highest expression in TNBC followed by the HER2^+^ subtype. The lowest PD-L1 expression is found in hormone-receptor positive advanced BC (0–10%), followed by hormone-receptor positive early BC (9–45%). In non-metastatic HER2^+^ BC, PD-L1 expression is around 30–35%, decreasing to 9–15% in advanced HER2^+^ BC. In metastatic TNBC, PD-L1 expression is present in 30 to 40% of patients, reaching up to 35–60% in early TNBC [16,43,44,52].

The prognostic value of PD-L1 expression by IHC in BC has discordant results between several studies, partially owing to technical issues related to different antibody clones, cutoff points and scoring systems. While few studies demonstrated a good correlation between PD-L1 expression and clinical outcome, others identified PD-L1 as a biomarker for worse survival, or no association was found [30,53,54,55].

The first study that showed PD-L1 expression as an independent negative prognostic factor in BC was published by Muenst et al. in 2015 [55]. A pre-diluted rabbit-anti-human PD-L1 polyclonal antibody was used. PD-L1 expression was quantified by using the modified Histo-score (H-score). H-score was calculated by a semi-quantitative assessment of both the intensity of staining and the percentage of PD-L1-positive tumor and immune cells (range of score 0 to 300). Since PD-L1 is expressed on the cell membrane, as well as the endomembrane system, both membranous and cytoplasmic staining were considered positive. PD-L1 was expressed in 152 (23.4%) out of the 650 BC patients, and it was significantly associated with age, large tumor size, lymph node status, tumor grade, absence of estrogen receptor expression, HER2-positive status and high expression of the proliferation marker Ki-67 (all *p*-values < 0.05). In univariate analysis, PD-L1 was associated with worse overall survival (OS) (*p*-value < 0.0001) [55].

In a recent publication by Noske et al. [56], 1318 BC samples were analyzed to examine the prevalence of PD-L1 expression, employing SP263 antibody clones (cutoff of ≥ 1%). PD-L1 expression was described in almost 74% in ICP and 60% in BC TC. PD-L1-positive immune cells in TNBC were associated with a significantly better disease-free survival (HR = 0.50 [0.25–0.99], *p*-value =0.0457). PD-L1 expression had no impact on patient outcome. The tumor-infiltrating lymphocytes (TILs) density was significantly associated with the expression of PD-1 and PD-L1 in immune cells (each *p*-value < 0.0001) and PD-L1 in tumor cells (*p*-value = 0.0051) [56].

In another study published in 2016 by Baptista et al., where 196 early BC cases were analyzed, they found a PD-L1 expression in 56.6% of BC samples employing 28-8 clones and TPS (PD-L1-positive if TPS ≥ 1), being significantly lower in ER-positive BC patients (*p*-value < 0.01). Moreover, PD-L1 expression was significantly associated with better OS (*p*-value = 0.04), despite its association with poor clinical and pathologic features [16].

Sabatier et al. [57] retrospectively analyzed PD-L1 mRNA expression in 45 BC cell lines and 5454 BC samples, using DNA microarrays. Compared to normal breast tissue, PD-L1 expression was found to be upregulated in about 20% of clinical samples and in up to 38% of TNBC. High PD-L1 expression was associated with worse clinicopathological parameters (large tumor size, high grade, high proliferation, ER-negative, PR-negative, HER2-positive status and basal tumors) (all *p*-values < 0.05). PDL1 upregulation was not associated with survival in the whole population, but was associated with better response to chemotherapy and overall specific survival in TNBC subtype (HR = 0.52 [0.38–0.71]; *p*-value = 0.00078) [49].

A meta-analysis by Wang et al. [52], evaluating nine relevant studies with 8583 patients, analyzed the prognostic value of PD-L1 in BC. However, significant heterogeneity was found in the number of patients included in each study, in the stage of the disease and in the clones and scores used for PD-L1 assessment. PD-L1 overexpression had no significant impact on metastasis-free survival, disease-free survival and overall specific survival, but it was significantly correlated with shortened OS (*p*-value = 0.045) [52].

A recent study from Van Berckelaer et al. [58] that included 143 non-pretreated patients with BC analyzed the prognostic value of PD-L1. PD-L1 expression was assessed by using SP142 assay on the TC and IC. Scoring was based on the percentage of the tumor area that was occupied by the percentage of PD-L1-positive tumor cells or PD-L1-positive immune cells. A score of 0 (PD-L1-negative), 1, 2 or 3 was assigned for <1%, ≥1% but <5%, ≥5% but <10% or ≥10% PD-L1-positive cells per tumor area, respectively. There was a strong correlation between PD-L1 positivity and TILs scores (*p*-value < 0.001). They observed that PD-L1 expression was correlated to the response to neoadjuvant therapy, but no association with prognosis was found [58].

A meta-analysis of Huang et al. [59] published in 2019 that included 47 studies with a total of 14,367 BC patients evaluated the association between PD-L1 expression and clinicopathological characteristics and BC prognosis. Various antibody clones were used to analyze PD-L1 in the included studies. PD-L1 expression evaluated by IHC in TC was associated with multiple high-risk factors, such as ductal carcinomas, large tumor size, histological grade 3 tumors, high Ki-67 and TNBC (all *p*-values < 0.05). Moreover, PD-L1-positive patients (TC ≥ 1%) were significantly associated with shorter disease-free survival (*p*-value < 0.0001) and OS (*p*-value = 0.006). Nevertheless, an exploratory analysis revealed that patients with PD-L1 overexpression, together with high tumor-infiltrating lymphocytes, may serve as a novel indicator for favorable prognosis with better OS (*p*-value < 0.0001) [59].

In 2021, Parvathareddy et al. [60] published their results from 1003 unselected Middle Eastern BC patients. Using a cutoff of ≥5%, they detected PD-L1 expression in TC, using E1L3N clones, in 32.8% of cases. PD-L1 was associated with worse clinicopathological parameters, such as younger age (*p*-value = 0.0432), higher grade (*p*-value = 0.0025), a high Ki-67 (*p*-value < 0.0001), ER-negative (*p*-value < 0.0001), PR-negative (*p*-value = 0.0001) and TNBC *(p*-value = 0.0062). A significant association between PD-L1 expression and deficient mismatch repair protein expression was found (*p*-value = 0.0009). However, there was no significant association between PD-L1 expression and OS (*p*-value = 0.6274). Nevertheless, upon further subgroup analysis, PD-L1 expression was correlated to better recurrence-free survival (*p*-value = 0.0043) and OS in TNBC (*p*-value = 0.0043) on multivariate analysis [60].

Further standardization of PD-L1 assessment assay and well-controlled clinical trials are warranted to clarify its prognostic value in BC.

PD-1/PD-L1 inhibitors have demonstrated great activity in the first clinical studies in BC, and some trials have tested their safety and efficacy in the neo/adjuvant and metastatic landscape. Actually, there are different therapeutic options in the BC setting. Pembrolizumab and nivolumab are monoclonal IgG antibodies with a high selectivity and affinity against PD-1, while atezolizumab and avelumab are monoclonal antibodies against PD-L1 [47].

### 4.3. PD-L1 and Immunotherapy in Hormone-Receptor Positive/HER2 Negative Breast Cancer

In recent studies evaluating patients with early stage hormone-receptor-positive and HER2-negative BC, PD-L1 expression was reported at around 9% in luminal A subtype and was increased to about 42% in luminal B [61]. In the metastatic stage, PD-L1 expression decreased significantly, being 0–1% in luminal A and 10–12% in luminal B patients [61].

Despite the few studies with immunotherapy in this subgroup (Table 3), data from initial phases of some clinical trials are reviewed in the following paragraphs.

The phase II GIADA trial [62] included 43 luminal B patients with stage II and IIIA. A neoadjuvant therapy of anthracycline-based chemotherapy and nivolumab plus endocrine treatment achieved a 16.3% of pathological complete response (pCR) rate, without reaching the primary endpoint [62]. Patients with T0/T1 with pCR vs. non-pCR obtained higher CD3+/PD-1+ and cytotoxic T-cell (CD8) expression (*p*-value = 0.01 and *p*-value = 0.001, respectively).

In the JAVELIN Solid Tumor phase Ib study [63], a total of 168 patients with metastatic BC who were refractory or progressed after standard treatment received avelumab [64]. Patients had been treated with a median of 3 prior lines for locally advanced or metastatic disease. A total of 43% of patients had hormone-receptor-positive/HER2-negative disease. The confirmed objective response rate (ORR) was 2.8% in this subset [63].

In the KEYNOTE-028 trial [65], 25 patients with highly pretreated ER+/HER2- advanced BC with a CPS of PD-L1 ≥ 1 (22C3 clone) received monotherapy with pembrolizumab. The ORR in this population was 12%, with a clinical benefit rate of 20%, and a modest but durable response in some patients [65].

### 4.4. PD-L1 and Immunotherapy in HER2 Positive Breast Cancer

PD-L1 is expressed in around 30% of early stage HER2+ BC tumors in humans, whereas it is detected in 9–10% of the tumor cells in the metastatic setting [61].

Some studies found that the expression of PD-1/PD-L1 in metastatic HER2+ BC was associated with poor outcomes, while no relation to clinical pathological features was described in primary tumors [66]. However, high levels of PD-1/PD-L1 expression with a high percentage of TILs in the tumor microenvironment were correlated with improved OS [67,68,69].

Hou et al. [70] assessed the relation of PD-L1 and other relevant immune biomarkers with clinical outcomes in a cohort of 123 early HER2^+^ BC patients. Among these cases, 64 had anti-HER2 neoadjuvant therapy, followed by surgical resection. Chemotherapy and anti-HER2 blockade were included in the treatment of all patients. PD-L1 expression evaluated by SP263 clones was identified in 72% of the patients (17% in tumor cells and 55% in cells of the immune system). Expression of PD-L1 was associated with high grade (*p*-value = 0.001), as well as a high level of CD8^+^ (*p*-value = 0.0002) and CD163+ cells (*p*-value = 0.0001). Thirty-nine out of 64 patients who received neoadjuvant treatment achieved pCR. They found that the negativity of progesterone receptor, intratumoral CD8^+^ cells and HER2/chromosome 17 centromere ratio were significantly associated with pCR (all *p*-values < 0.05). Furthermore, all patients who expressed intratumoral CD8^+^ cells but no PD-L1 positivity reached pCR. Based on their findings, the authors suggested that the analysis of CD8+ cells in the tumor in conjunction with PD-L1 expression could be useful in prognosticating the response to anti-HER2 treatment in patients with HER2^+^ BC [70].

The effectiveness of anti-PD-1/PD-L1 antibodies in combination with HER2-targeted therapy to increase the efficacy of BC treatment is a subject that is under investigation (Table 3). Recently, the phase II randomized KATE2 trial [71] assessed the combination of atezolizumab to ado-trastuzumab emtansine (T-DM1) in 202 patients diagnosed with locally advanced or metastatic HER2+ BC patients who had received prior treatment with trastuzumab and taxane-based chemotherapy. Patients were stratified based on PD-L1 expression (SP142 clone, IC ≥ 1%,), 41.6% of which were PD-L1-enrichment. In the overall population, the OS at 1 year was similar in both groups. In the PD-L1-positive patients, 1-year OS was higher in the atezolizumab + T-DM1 arm (94.3% vs. 87.9%). Even though the follow-up time is short, these data indicate a possible benefit in terms of OS with atezolizumab + T-DM1 in PD-L1-positive patients [71].

The combination of pembrolizumab plus trastuzumab in patients with HER2+ advanced BC who had progressed to therapy with trastuzumab was explored in the phase Ib/II PANACEA trial [72]. A total of 77% of the patients included in the trial presented PD-L1 expression by 22C3 assay (CPS ≥ 1). The objective response was 15% in PD-L1-positive patients, without objective responders among the PD-L1-negative patients [72].

In the previously exposed JAVELIN Solid Tumor phase Ib trial [63], 15.5% of the patients presented HER2-positive disease. No ORR was observed in this heavily pretreated subgroup of patients.

### 4.5. PD-L1 Expression and Immunotherapy in Triple-Negative Breast Cancer

In early TNBC, previous studies have found that PD-L1 is overexpressed in around 45 to 55% of the tumor cells, whereas, in the advanced disease, the expression of PD-L1 is about 35% [61,73].

Some studies have examined the expression of PD-L1 and PD-L2 in patients with early TNBC, showing that about 55% and 50%, respectively, were positive [16]. Unexpectedly, despite a higher relapse rate in PD-L1-positive patients, their OS was better than in PD-L1-negative subgroups [57,73], a fact that could be linked to a stronger underlying antitumor immune response secondary to treatment [55]. Moreover, PD-L1 expression in TNBC has been positively associated with the expression of other immune system regulators, such as indoleamine 2,3-dioxygenase 1 (IDO1) and cytotoxic T-lymphocyte antigen 4 (CTLA-4) in addition to *BRCA1* gene mutations [17].

Despite their disagreement regarding the absolute concentrations of PD-L1 reported in TNBC, data from the previously referred clinical studies support the immunotherapy as a promising treatment approach for TNBC [74].

Studies with neoadjuvant chemotherapy plus immunotherapy with anti-PD-1/L1 in early TNBC settings have been presented recently (Table 3). The GeparNuevo trial is a phase II randomized study that included 174 patients to receive durvalumab versus a placebo administered together with nab-paclitaxel, followed by epirubicin and cyclophosphamide [75]. In the window phase, durvalumab/placebo was delivered 2 weeks before the initiation of nab-paclitaxel. Up to 53.4% of the patients in the durvalumab arm achieved a pCR vs. 44.2% in the placebo group, without reaching the limit of statistical significance. A total of 87% of the patients received PD-L1-enrichment by using SP263 clones (TC ≥ 1% and/or IC ≥ 1%). A trend towards an increase in pCR rates was observed in PD-L1-positive patients and was statistically significant for both the durvalumab (*p*-value = 0.045) and the placebo arm (*p*-value = 0.040) [75].

In the phase III KEYNOTE-522 study [76], a total of 602 stage II or III TNBC patients were randomized to receive neoadjuvant treatment with paclitaxel and carboplatin plus pembrolizumab or placebo. The two subgroups of patients were subsequently treated with four additional cycles of anthracyclines plus cyclophosphamide, together with pembrolizumab or a placebo. After surgery, all the patients received adjuvant pembrolizumab versus a placebo every 3 weeks for up to nine cycles. In this early stage cohort, approximately 80% of the intention-to-treat population was considered PD-L1-positive (22C3 clone, CPS ≥ 1). The percentage of patients with a pCR was 64.8% in the pembrolizumab–chemotherapy group and 51.2% in the chemotherapy group (*p*-value < 0.001). In this trial, a significant increase of 13.6% in the pCR rate was noted in the experimental arm when pembrolizumab was administered with chemotherapy. The benefit of the combination was independent of PD-L1 expression, getting PD-L1-positive patients a higher pCR in both arms compared with the PD-L1-negative population [76].

In spite of the encouraging results observed in the previous trial, the phase III NeoTRIP [77] study did not find a significant difference in the pCR with the addition of atezolizumab to nab-paclitaxel and carboplatin (43.5% vs. 40.8%, *p*-value = 0.066) in a very similar population of BC patients [77,78,79]. However, it is important to note that the principal objective in the NeoTRIP trial was the evaluation of disease-free survival (not yet reached), instead of the pCR, as in GeparNuevo and KEYNOTE-522 studies. Moreover, the neoadjuvant chemotherapy strategy was different between these trials, excluding the use of cyclophosphamide and anthracyclines in the NeoTRIP trial, which are considered quite immunogenic agents.

Recently, the results from the phase III Impassion031 study were published [80]. A total of 333 patients with early or locally advanced TNBC were randomized to receive treatment with nab-paclitaxel, followed by cyclophosphamide plus doxorubicin and atezolizumab or a placebo. After definitive surgery, 11 cycles of atezolizumab were administered in the experimental group. Moreover, pCR was obtained in 57.6% of the patients in the combination arm versus 41% of the patients in the control group (*p*-value = 0.0044; significance boundary ≤ 0.0184). In total, 46% of patients were PD-L1 enriched. PD-L1 evaluation was measured through SP142 assay, with PD-L1 positive defined as IC ≥ 1%. In the PD-L1-positive patients, pCR was obtained in 68.8% of the patients in the immunotherapy group vs. 49.3% in the placebo arm (*p*-value = 0.021; significance boundary ≤ 0.0184) [74,80].

Actually, a few studies with anti-PD-1/L1 agents have been published in the metastatic setting (Table 3). In the JAVELIN Solid Tumor phase Ib trial [63], patients with metastatic BC who were refractory or progressed after standard therapy received treatment with avelumab. The ORR was 3% overall, and it was 5.2% in patients with TNBC, with the disease control rate being 28% vs. 31%, respectively. In addition, a trend towards a greater ORR was noted both in patients with PD-L1-positive and PD-L1-negative in the tumor immune cells, using SP263 clones, in the global cohort (16.7% vs. 1.6%) and in the TNBC subtype (22.2% vs. 2.6%) [63].

In the single-arm phase Ib KEYNOTE-012 [81] clinical trial, treatment with pembrolizumab was administered in PD-L1-expressing recurrent or metastatic TNBC. A total of 58% of 111 patients screened had PD-L1 overexpression (22C3 clone, CPS ≥ 1). The ORR was 18.5% in this pretreated population [81].

In the recent KEYNOTE-119 [78] phase III trial, a total of 622 pretreated patients were randomized to receive pembrolizumab monotherapy vs. single-drug chemotherapy per investigator’s choice. Randomization was stratified according to PD-L1 (22C3 clone) tumor status (considered positive if CPS ≥ 1 and negative if CPS < 1) and previous history of neo/adjuvant therapy vs. de novo metastatic disease. The median OS in patients with PD-L1 ≥ 10 measured by CPS was 12.7 months in the pembrolizumab arm versus 11.6 months in the chemotherapy group (*p*-value = 0.057). In participants with a CPS ≥ 1, the median OS was 10.7 vs. 10.2 months, respectively (*p*-value = 0.073). In the global population, the median OS was also similar in both subgroups (9.9 months in the immunotherapy and 10.8 months in the chemotherapy arm). The authors concluded that treatment with pembrolizumab did not significantly increase OS over chemotherapy in patients with previously treated metastatic TNBC, regardless of the expression of PD-L1 [78].

The KEYNOTE-355 study evaluated the treatment with pembrolizumab in patients with recurrent inoperable or metastatic TNBC in the first-line setting [82]. During this trial, 847 patients were randomized 2:1 to receive chemotherapy in addition to pembrolizumab versus placebo. Randomization was stratified by sort of chemotherapy, basal expression of PD-L1 by 22C3 clone (CPS ≥ 1 or < 1), and previous therapy within the neo/adjuvant setting. In total, 25% of the patients presented PD-L1 CPS < 1, 75% CPS ≥ 1, and 38% CPS ≥ Data of progression-free survival (PFS) are available, still pending the results of the OS. In the subgroup of patients with CPS ≥ 10, the median PFS was significantly higher in the combination group (9.7 vs. 5.6 months) (*p*-value = 0.0012, significance boundary ≤ 0.00411). The PFS rate at 6 and 12 months was higher in the pembrolizumab–chemotherapy combination vs. chemotherapy alone group in patients with CPS ≥ 1 (56.4% vs. 46.6% and 31.7% vs. 19.4%, respectively) (*p*-value = 0.0014; significance boundary ≤ 0.00111) [82]. However, no differences in median PFS were found in patients with PD-L1 CPS < 1 in both treatment arms (6.3 vs. 6.2 months), contrary to the benefit seen with the combination strategy in all patients in the early setting.

Recently, the phase III IMpassion130 clinical trial [36], including 902 patients, also explored whether the addition of atezolizumab to nab-paclitaxel chemotherapy could improve the outcomes in TNBC patients in the first-line treatment. PD-L1 was assessed by SP142 assay and immune cell score (PD-L1 positive if IC ≥ 1%). The basal expression of PD-L1 was considered a stratification factor, being 41% of the patients PD-L1-positive. In the general population, the median PFS was higher in the combination group, but this finding did not correlate with a significant benefit in OS (21 vs. 18.7 months respectively) (*p*-value = 0.077). In contrast, clinically meaningful OS improvement was noted in PD-L1-positive patients (25.4 vs. 17.9 months) with a hazard ratio of 0.67 (95%CI: 0.53–0.86) [83]. For this reason, atezolizumab plus nab-paclitaxel is a valid first-line option for PD-L1-positive metastatic TNBC, being approved in the United States and Europe for this indication. This admission, however, is linked to the detection of PD-L1-positive immune cells with an IC score of at least 1%. Because of this, oncologists should order PD-L1 testing with the information of which immunotherapy they plan to use.

The IMpassion131 trial [84] also evaluated the combination of atezolizumab with paclitaxel in patients with TNBC in the first-line setting. The addition of this immunotherapy to paclitaxel did not improve PFS in PD-L1-positive (SP142, IC ≥ 1%) patients or in the overall population. No subgroup of patients had additional benefits from the checkpoint inhibitor. The possible reasons for the different outcomes observed in this study in contrast with the IMpassion130 trial require further investigations.

According to the results of these studies, the combination strategy of chemotherapy and immunotherapy could be a new standard of care in early TNBC, and also in advanced disease for the subgroup of patients with PD-L1-positive (IC score of ≥ 1%, using the SP142 assay for atezolizumab, or CPS of ≥ 10, using the 22C3 assay for pembrolizumab). However, different methods for the assessment of PD-L1 were used between trials, which could have contributed to the disparity of the results.

There are some possible biological explanations to justify why immunotherapy can benefit TNBC patients regardless of PD-L1 positivity in localized disease, but not in the metastatic setting. In the first place, it is important to consider that treatment is more effective in the early disease, and, hence, new tumor antigens can be created. Moreover, the host immune system is probably more robust in this scenario due to the limited cancer burden and the major effectiveness in triggering an antitumor immunologic response to new antigens.

The TNBC phenotype is very heterogeneous and has different biological characteristics. In the last few years, a great effort has been made in order to better characterize the diverse types of TNBC based on their somatic mutational profile to find possible therapeutic targets [85,86]. The immunomodulatory TNBC subtype shows an increase in the expression of genes involved in immune signaling pathways, including the PD-1/PD-L1 pathway [87], with immunotherapy being a good treatment option in this subgroup.

At the present time, for PD-L1 positive advanced TNBC patients, the preferred option is chemotherapy and immunotherapy combination. In the case of PD-L1 positivity assessed by SP142, the new standard is nab-paclitaxel plus atezolizumab. In the case of PD-L1 CPS ≥ 10 measured by 22C3, the treatment consists of chemotherapy plus pembrolizumab. Other treatment options, such as olaparib, talazoparib or chemotherapy with carboplatin for BRCA mutated patients, as well as chemotherapy with or without bevacizumab, for PD-L1 negative patients are available in the metastatic TNBC field [88].

## 5. Conclusions

PD-L1 is becoming an emerging biomarker in breast cancer, following the path of other more immunogenic tumors. Usually, the IHC technique is used to assess PD-L1 expression, but there are different scores, antibodies clones and PD-L1 cutoff points for its analysis. Performing PD-L1 expression in BC is recommended in all patients, especially in the triple-negative subtype. Despite studies that have shown divergent results, PD-L1 has been correlated with worse clinicopathological parameters in BC and poor outcomes in patients with hormone receptor positive. However, PD-L1 expression seems to have a favorable impact on OS in the TNBC subtype.

Moreover, PD-L1 is emerging as a predictive biomarker to guide the response to systemic treatment in TNBC. In the near future, immunotherapy will be part of the neoadjuvant treatment in TNBC, where all patients appear to benefit regardless of PD-L1 expression. Immune checkpoint inhibitors plus chemotherapy are a new treatment option in the metastatic setting in patients with positive PD-L1 expression. In advanced TNBC, performing PD-L1 analysis should be mandatory to choose the best first-line strategy. The technique to assess PD-L1 in advanced TNBC should be decided based on the drug the oncologist is planning to use (IC employing the SP142 assay for atezolizumab or CPS employing the 22C3 assay for pembrolizumab). More studies are needed in order to clarify why immunotherapy benefits TNBC patients regardless of PD-L1 positivity in the localized disease, but not in the metastatic setting.

## Figures and Tables

**Figure 1 cancers-14-00307-f001:**
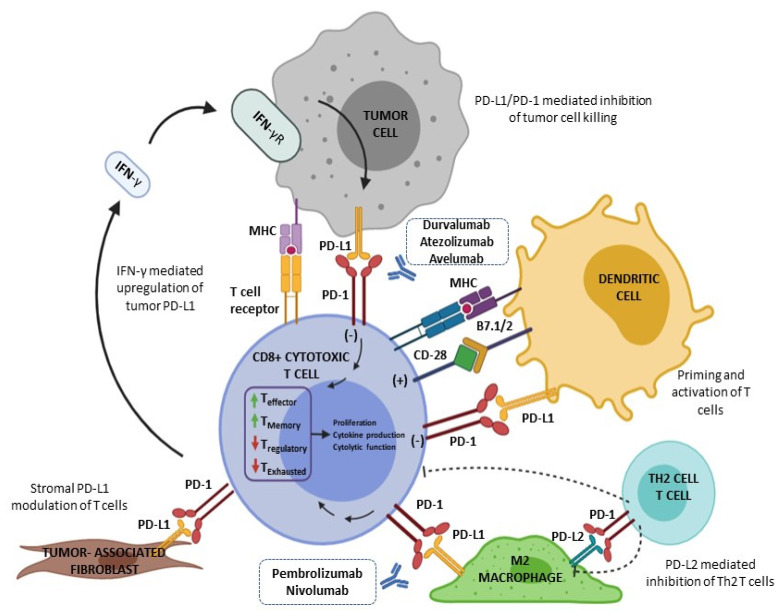
Mechanism of action of anti PD-1/L1. Activated T cells express PD-1, and when it binds to PD-L1/2 on tumor cells, it results in T-cell depletion. Durvalumab, avelumab and atezolizumab block PD-L1, while pembrolizumab and nivolumab block PD-1 to produce antitumor responses.

**Table 1 cancers-14-00307-t001:** Relevant PD-L1 scores assessment in malignant tumors.

PD-L1 Scores	Analysis Assessment	References
Tumor Cell (TC)	TC (%) = [Number of PD-L1-stained tumor cells/Total number of viable tumor cells] × 100%	[18,22,23]
Tumor-Proportion Score (TPS)	TPS (%) = [Number of PD-L1-stained tumor cells/Total number of viable tumor cell] × 100%	[18,22,24]
Immune-Cell Score (IC)	IC (%) = [Area of tumor infiltrated by PD-L1-stained immune cells/Total tumor area] × 100%	[18,22,25]
Immune Cells Present (ICP)	ICP (%) = Percentage of tumor area occupied by any PD-L1 positive immune cell staining.	[25]
Combined Positive Score (CPS)	CPS = [Number of PD-L1-stained cells (tumor cells, lymphocytes and macrophages)/Total number of viable tumor cells] × 100	[18,22,23]

**Table 2 cancers-14-00307-t002:** Most commonly used IHC assays in malignant tumors with the scoring methods and predictive drugs.

Antibody Clone	Platform	Scoring Method	Predictive Drug	References
SP142 (monoclonal, rabbit)	BenchMark Ultra	IC or TC	Atezolizumab (anti-PD-L1)	[5,6,18,27,28,29,30]
SP263 (monoclonal, rabbit)	BenchMark Ultra	TPS, TC or IC	Durvalumab (anti-PD-L1)	[5,6,18,27,28,29,30]
22C3 (monoclonal, mouse)	Dako Autostainer Link 48	TPS or CPS	Pembrolizumab (anti-PD-1)	[5,6,18,27,28,29,30]
28-8 (monoclonal, rabbit)	Dako Autostainer Link 48	TPS	Nivolumab (anti-PD-1)	[5,6,18,27,28,29,30]
E1L3N (monoclonal, rabbit)	Not linked to a specific staining platform	IC, TPS or CPS	Non-associated drug	[5,6,18,27,28,29,30,31]
73-10 (monoclonal, rabbit)	Dako Autostainer Link 48	Not established yet	Avelumab (anti-PD-L1)	[6,22]

IC, immune-cell score; TCs, tumor cells; TPS, Tumor-Proportion Score; CPS, Combined Positive Score.

**Table 3 cancers-14-00307-t003:** Summary of the most relevant studies with immunotherapy in the different subtypes of breast cancer.

Clinical Trial	Breast Cancer Subtype and Stage	Number of Patients Included	PD-L1 Expression	Antibody Clone	Immunotherapy Drug	Associated Drugs	Response Rates
GIADA	Early luminal B	43	Any expression	28-8 (DAKO PharmDx)	Nivolumab	Anthracyclines plus endocrine therapy	16.3% pCR
JAVELIN Solid Tumor	Pretreated and metastatic BC	168	Any expression	73–10 (DAKO PharmDx)	Avelumab	No other drugs	ORR 2.8% in luminal, 0% in HER2^+^ and 5.2% in TNBC
KEYNOTE-028	Pretreated and advanced luminal BC	25	CPS ≥ 1	22C3 (DAKO PharmDx)	Pembrolizumab	No other drugs	ORR 12%
KATE2	Pretreated and advanced HER2-positive BC	202	Any expression	SP142 (VENTANA)	Atezolizumab	T-DM1	Unknown ORR
PANACEA	Pretreated and advanced HER2-positive BC	58	Any expression	22C3 (DAKO PharmDx)	Pembrolizumab	Trastuzumab	ORR 15% in PD-L1+, no ORR in PD-L1-
GeparNuevo	Early TNBC	174	Any expression	SP263 (VENTANA)	Durvalumab vs. placebo	Nab-paclitaxel, + epirubicin and CP	53.4% of pCR in durvalumab arm vs. 44.2% in placebo arm
KEYNOTE-522	Early TNBC	602	Any expression	22C3 (DAKO PharmDx)	Pembrolizumab vs. placebo	Paclitaxel and carboplatin + anthracyclines and CP	64.8% of pCR in pembrolizumab arm vs. 51.2% in placebo arm
NeoTRIP	Early TNBC	280	Any expression	SP142 (VENTANA)	Atezolizumab	Nab-paclitaxel + carboplatin	43.5% of pCR in atezolizumab arm vs. 40.8% in placebo arm
IMpassion 031	Early TNBC	333	Any expression	SP142 (VENTANA)	Atezolizumab	Nab-paclitaxel + doxorubicin and CP	57.6% of pCR in atezolizumab arm vs. 41% in placebo arm
KEYNOTE-012	Pretreated and advanced TNBC	111	≥1% of TCs and IC by IHC	22C3 (DAKO PharmDx)	Pembrolizumab	No other drugs	ORR 18.5%
KEYNOTE-119	Pretreated and advanced TNBC	622	Any expression. Stratified by PD-L1	22C3 (DAKO PharmDx)	Pembrolizumab vs. single-drug CT	No other drugs	ORR (~9% in both arms)
KEYNOTE-355	Advanced TNBC	847	Any expression. Stratified by PD-L1	22C3 (DAKO PharmDx)	Pembrolizumab vs. placebo	Chemotherapy	ORR 41% in combination arm and 35.9% in CT arm
IMpassion 130	Advanced TNBC	902	Any expression. Stratified by PD-L1	SP142 (VENTANA)	Atezolizumab vs. placebo	Nab-paclitaxel	ORR 56% in combination arm and 45.9% in CT arm
IMpassion 131	Advanced TNBC	651	Any expression. Stratified by PD-L1	SP142 (VENTANA)	Atezolizumab vs. placebo	Paclitaxel	ORR 53.6% in combination arm and 47.5% in CT arm

Legend: BC, breast cancer; CP, cyclophosphamide; pCR, pathological complete response; ORR, Overall Response Rate; CT, chemotherapy.

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
