# Peer review of "Programmed Death-Ligand 1 (PD-L1) as Immunotherapy Biomarker in Breast Cancer"

_cancers, 2022, doi:10.3390/cancers14020307_

Round 1
Reviewer 1 Report
In the manuscript titled “Analysis of Programmed Death-ligand 1 (PD-L1) as a Predictive
and Prognostic Biomarker in Breast Cancer”, the authors have attempted to perform a literature review of the role of PD-L1 as a predictive/prognostic biomarker in breast cancer. This review article has some merit, since immunotherapy with PD-1/PD-L1 inhibitors is fast gaining ground as a therapeutic option in different tumors, with breast cancer being no exception. However, a few concerns need to be addressed by the authors, before the manuscript can be accepted.
- Although the stated objective and the title suggest that the review focuses on predictive/prognostic role of PD-L1 in breast cancer, a vast majority of the manuscript is dedicated to reporting the clinical trials data in different molecular subtypes of breast cancer. The title can be modified to better reflect the contents of the article.
- In the Introduction (Page 1) and Discussion (Page 3) sections, authors have erroneously referred to the DAKO antibody as DAKO 2C3 instead of DAKO 22C3.
- In Introduction section (Page 1), “…. with the highest expression in triple negative breast cancer (TNBC).” Please add a reference.
- In Discussion section (Page 2), “The function of PD-L2 is less known, and its clinical role is the scope of current investigation (9).” Although the authors claim that the clinical role of PD-L2 is the scope of current investigation, the reference is from 2001. Please add some recent references or modify the statement.
- In section 4.1 (Page 4), the authors state “PD-L1 expression by IHC in BC is low (around 10-30%) compared with other tumors such as lung cancer, and varies with stage and molecular subtype, with the highest expression in TNBC followed by the HER2+ subtype (12) (Table 2).” The authors have generalized the frequency of PD-L1 expression in different stages and molecular subtypes of breast cancer based on a single reference. This is not acceptable. Please modify the statement as well as table 2 based on multiple references.
- In section 4.1 (Page 4 and 5), more references need to be added and more studies need to be discussed with regards to the predictive/prognostic role of PD-L1 in different molecular subtypes of breast cancer.
- In Table 3 (Page 8), it will be beneficial to the readers to see which PD-L1 antibody was used in each clinical trial listed, to assess the PD-L1 expression.
- The manuscript needs extensive English language editing by a native English speaker.
Author Response
REVIEWER 1
- First of all, we would like to thank you for your corrections. Following your recommendations, we have now prepared a thoroughly revised version of the manuscript, in which we have addressed all the concerns.
We sincerely appreciate the comments and believe these revisions will improve the quality of our manuscript.
In the manuscript titled “Analysis of Programmed Death-ligand 1 (PD-L1) as a Predictive and Prognostic Biomarker in Breast Cancer”, the authors have attempted to perform a literature review of the role of PD-L1 as a predictive/prognostic biomarker in breast cancer. This review article has some merit, since immunotherapy with PD-1/PD-L1 inhibitors is fast gaining ground as a therapeutic option in different tumors, with breast cancer being no exception. However, a few concerns need to be addressed by the authors, before the manuscript can be accepted.
Although the stated objective and the title suggest that the review focuses on the predictive/prognostic role of PD-L1 in breast cancer, a vast majority of the manuscript is dedicated to reporting the clinical trials data in different molecular subtypes of breast cancer. The title can be modified to better reflect the contents of the article.
-
We are grateful for this comment. As the reviewer suggested, we have modified the title to include the clinical trials using immunotherapy in breast cancer.
In the Introduction (Page 1) and Discussion (Page 3) sections, authors have erroneously referred to the DAKO antibody as DAKO 2C3 instead of DAKO 22C3.
-
We have corrected the typographic error about DAKO 22C3 (lines 62 and 136).
In the Introduction section (Page 1), “…. with the highest expression in triple negative breast cancer (TNBC).” Please add a reference.
-
In line 70 we have added two references about the highest expression of PD-L1 in TNBC. The references are:
Hong M, Kim JW, Kim M kyoon, Chung B wha, Ahn S kyung. Programmed cell death-ligand 1 expression in stromal immune cells is a marker of breast cancer outcome. J Cancer. 2020;11(24):7246–52.
Zhang M, Sun H, Zhao S, Wang Y, Pu H, Zhang Q. Expression of PD-L1 and prognosis in breast cancer: A metaanalysis. Oncotarget. 2017;8(19):31347–54.
-
We also added more references in the subsection: 4.5. PD-L1 expression and immunotherapy in triple-negative breast cancer.
-
Lines 317 and 321. The references are:
Buisseret L, Garaud S, de Wind A, Van den Eynden G, Boisson A, Solinas C, Gu-Trantien C, Naveaux C, Lodewyckx JN, Duvillier H, Craciun L, Veys I, Larsimont D, Piccart-Gebhart M, Stagg J, Sotiriou C, Willard-Gallo K. Tumor-infiltrating lymphocyte composition, organization and PD-1/ PD-L1 expression are linked in breast cancer. Oncoimmunology. 2016 Dec 14; 6(1):e1257452.
Ghosh J, Chatterjee M, Ganguly S, Datta A, Biswas B, Mukherjee G, Agarwal S, Ahmed R, Chatterjee S, Dabkara D. PDL1 expression and its correlation with outcomes in non-metastatic triple-negative breast cancer (TNBC). Ecancermedicalscience. 2021 Apr 6; 15:1217.
In Discussion section (Page 2), “The function of PD-L2 is less known, and its clinical role is the scope of current investigation (9).” Although the authors claim that the clinical role of PD-L2 is the scope of current investigation, the reference is from 2001. Please add some recent references or modify the statement.
-
We have added more information about PD-L2 and new references.
-
Lines 121-125: The function of PD-L2 is less known, being principally an inhibitory molecule expressed not only by antigen-presenting cells, but also by other immune cells in an inducible manner, mainly through Th2-associated cytokines [12]. Its clinical role is the scope of current investigation [13–15].
-
References:
Rozali EN, Hato S V., Robinson BW, Lake RA, Lesterhuis WJ. Programmed death ligand 2 in cancer-induced immune suppression. Clin Dev Immunol. 2012;2012:656340
Latchman Y, Wood CR, Chernova T, Chaudhary D, Borde M, Chernova I, Iwai Y, Long AJ, Brown JA, Nunes R, et al. PD-L2 is a second ligand for PD-1 and inhibits T cell activation. Nat Immunol. 2001;2(3):261–8.
Baptista MZ, Sarian LO, Derchain SFM, Pinto GA, Vassallo J. Prognostic significance of PD-L1 and PD-L2 in breast cancer. Hum Pathol. 2016;47(1):78–84.
Fang J, Chen F, Liu D, Gu F, Chen Z, Wang Y. Prognostic value of immune checkpoint molecules in breast cancer. Biosci Rep. 2020 Jul 31;40(7):BSR20201054.
In section 4.1 (Page 4), the authors state “PD-L1 expression by IHC in BC is low (around 10-30%) compared with other tumors such as lung cancer, and varies with stage and molecular subtype, with the highest expression in TNBC followed by the HER2+ subtype (12) (Table 2).” The authors have generalized the frequency of PD-L1 expression in different stages and molecular subtypes of breast cancer based on a single reference. This is not acceptable. Please modify the statement as well as table 2 based on multiple references.
-
We have added more references of different studies, although bibliography is limited. We have eliminated table 2 to avoid confusions for readers. Table 2 information was remade in a new paragraph.
-
Lines 187-200: PD-L1 expression by IHC in BC is low (10-30%) compared with other tumors such as non-small cell lung cancer (around 70%) [28], and varies with stage and molecular subtype, with the highest expression in TNBC followed by the HER2+ subtype. The lowest PD-L1 expression is found in hormone receptor positive advanced BC (0-10%) followed by hormone receptor positive early BC (9-45%). In non-metastatic HER2+ BC, PD-L1 expression is around 30-35% decreasing to 9-15% in advanced HER2+ BC. In metastatic TNBC, PD-L1 expression is present in 30 to 40% of patients, reaching up to 35-60% in early TNBC [14,20,21,29,30]. The prognostic value of PD-L1 expression by IHC in BC has discordant results between several studies, partially owing to technical issues related to different antibody clones, cutoff points, and scoring systems. While some studies demonstrated a good correlation between PD-L1 expression and clinical outcome, others identified PD-L1 as a biomarker for worse survival, or no association was found [19,31–33].
-
The references are:
Baptista MZ, Sarian LO, Derchain SFM, Pinto GA, Vassallo J. Prognostic significance of PD-L1 and PD-L2 in breast cancer. Hum Pathol. 2016;47(1):78–84.
Guo H, Ding Q, Gong Y, Gilcrease MZ, Zhao M, Zhao J, Sui D, Wu Y, Chen H, Liu H, et al. Comparison of three scoring methods using the FDA-approved 22C3 immunohistochemistry assay to evaluate PD-L1 expression in breast cancer and their association with clinicopathologic factors. Breast Cancer Res. 2020;22(1):1–18.
Ohaegbulam KC, Assal A, Lazar-Molnar E, Yao Y, Zang X. Human cancer immunotherapy with antibodies to the PD-1 and PD-L1 pathway. Trends Mol Med. 2015;21(1):24–33.
Allison JP. Immune checkpoint blockade in cancer therapy the 2015 Lasker-Debakey clinical medical research award. JAMA - J Am Med Assoc. 2015;314(11):1113–4.
Parvathareddy SK, Siraj AK, Ahmed SO, Ghazwani LO, Aldughaither SM, Al-Dayel F, et al. PBaptista MZ, Sarian LO, Derchain SFM, Pinto GA, Vassallo J. Prognostic significance of PD-L1 and PD-L2 in breast cancer. Hum Pathol. 2016; Cells. 2021;10(2).
Wang C, Zhu H, Zhou Y, Mao F, Lin Y, Pan B, Zhang X, Xu Q, Huang X, Sun Q. Prognostic Value of PD-L1 in Breast Cancer: A Meta-Analysis. Breast J. 2017;23(4):436–43.
In section 4.1 (Page 4 and 5), more references need to be added and more studies need to be discussed with regards to the predictive/prognostic role of PD-L1 in different molecular subtypes of breast cancer.
-
We have introduced several changes and added some meta-analysis and studies to discuss the predictive/prognostic role of PD-L1. Next paragraphs were added to the review:
-
Lines 196-200: The prognostic value of PD-L1 expression by IHC in BC has discordant results between several studies, partially owing to technical issues related to different antibody clones, cutoff points, and scoring systems. While some studies demonstrated a good correlation between PD-L1 expression and clinical outcome, others identified PD-L1 as a biomarker for worse survival, or no association was found [19,31–33].
-
Lines 222-224: A first meta-analysis by Wang et al. [30] evaluating 9 relevant but heterogeneous studies analyzed the prognostic value of PD-L1 in BC. The results showed that PD-L1 expression had no significant impact on disease free survival and overall specific survival, but shorter OS [30].
A recent study from Van Berckelaer et al. [36] that included 143 non-pretreated patients with BC analyzed the prognostic value of PD-L1. PD-L1 expression was assessed using a PD-L1 (SP142) assay. They observed that PD-L1 expression was correlated to response to neoadjuvant therapy but no association with prognosis was found [36].
A meta-analysis of Huang et al. [37] published in 2019 included 47 studies. PD-L1 expression evaluated by IHC in tumor cells was associated with multiple high-risk factors, such as ductal carcinomas, large tumor size, histological grade 3 tumors, high Ki-67, TNBC, and shorter OS. Nevertheless, an exploratory analysis revealed that patients with PD-L1 overexpression together with high tumor infiltrating lymphocytes may serve as a novel indicator for favorable prognosis with better OS [37].
In 2021, Parvathareddy et al. [29] published their results from 1,003 unselected Middle Eastern BC patients. PD-L1 expression was detected in 32.8% of cases. No association was found between PD-L1 expression and clinical outcome. However, on further subgroup analysis, PD-L1 expression was correlated to recurrence-free survival and OS in TNBC [29].
Further standardization of PD-L1 assessment assay and well-controlled clinical trials are warranted to clarify its prognostic value in BC.
In Table 3 (Page 8), it will be beneficial to the readers to see which PD-L1 antibody was used in each clinical trial listed, to assess the PD-L1 expression.
-
As the reviewer suggested, in Page 11-12, we have added which PD-L1 antibody was used in each clinical trial listed in the new Table 2.
The manuscript needs extensive English language editing by a native English speaker.
-
The text has been proofread by native English speaker. Moreover, we have corrected some typographical errors in the text, and redacted the bibliography according to the journal system.
Reviewer 2 Report
General comment
The work summarizes and discusses the latest clinical trials assessing the benefits of PD-1/PD-L1 immunotherapy in breast cancer. The authors attempted to decipher the causes of differences between the trial outcomes, which are contrasting in many cases. The work provides a structured update on the status of PD-1/PD-L1 in breast cancer. The work is well structured and systematically provides a general overview of the field and, therefore, should be of interest to the readers.
Minor concerns:
1/ The text is acceptably readable, but there are mistakes and typo errors. The text needs English editing.
2/ It would be helpful to graphically show the basics of the PD-1/PD-L1/2 signaling since the whole review revolves around this.
3/ The authors point out that IHC assays use different antibodies for their analyses, which might be a reason for differences between them. The authors should elaborate on which antibodies (clones, etc.) are used for the IHC scoring assays.
Author Response
REVIEWER 2
- First of all, we would like to thank you for your corrections. Following your recommendations, we have now prepared a thoroughly revised version of the manuscript, in which we have addressed all the concerns.
We sincerely appreciate the comments and believe these revisions will improve the quality of our manuscript.
The work summarizes and discusses the latest clinical trials assessing the benefits of PD-1/PD-L1 immunotherapy in breast cancer. The authors attempted to decipher the causes of differences between the trial outcomes, which are contrasting in many cases. The work provides a structured update on the status of PD-1/PD-L1 in breast cancer. The work is well structured and systematically provides a general overview of the field and, therefore, should be of interest to the readers.
Minor concerns:
1/ The text is acceptably readable, but there are mistakes and typo errors. The text needs English editing.
-
The text has been proofread by native English speaker. Moreover, we have corrected some typographical errors in the text, and redacted the bibliography according to the journal system.
2/ It would be helpful to graphically show the basics of the PD-1/PD-L1/2 signaling since the whole review revolves around this.
-
As the reviewer suggested, we have improved Figure 1 (page 4), showing all the PD-1/PD-L1/2 signaling, including the intra-cell pathway.
3/ The authors point out that IHC assays use different antibodies for their analyses, which might be a reason for differences between them. The authors should elaborate on which
-
In the review, lines 131-149; we discuss the four clinically developed PD-L1 assays and their differences, including the different ways to analyze PD-L1. Also in Table 1 (page 5), we show the diverse antibody clones with the analyzed tissue and the corresponding predictive drugs.
-
Moreover, we have added more information, a new paragraph and more references to discuss the differences between them.
-
Lines 131-149: The present PD-L1 immunohistochemistry (IHC) biomarker landscape is complex. Various IHC assays with heterogeneous scoring algorithms are approved for different therapies and tumor indications [3]. Currently, there are four clinically developed PD-L1 assays evaluating tumor proportion score (TPS), immune cell score (IC), presence of tumor-associated immune cells (ICP) and combined positive score (CPS): SP142, SP263, 28-82 and 22C3 (table 1). In the measurement of PD-L1 by TPS, just membranous staining of tumor cells is considered as significant. On the other hand, IC is restricted to PD-L1 expression in some inflammatory cells (T-cells, macrophages, among others), and CPS is calculated based on the number of PD-L1 positive cells (lymphocytes, macrophages and tumor) respecting total tumor cells [16].The most frequently IHC assays used to determine the expression level of PD-L1 are SP142 Assay (PD-L1 SP142, VENTANA) and PD-L1 IHC 22C3pharmDx Assay (PD-L1 22C3, DAKO). PD-L1 SP142 was used for the approval of atezolizumab, whereas PD-L1 22C3 is predictive for pembrolizumab. PD-L1-positive expression is considered when the tumor proportion score (TPS) is 1% or higher, or the combined positive score (CPS) is ≥ 1 [3].
Some studies have tried to correlate the concordance between diverse PD-L1 IHC assays in different tumors, including BC, showing divergent data. For this reason, they cannot be used interchangeably in clinical practice [17–19].
-
The references are:
-
Huang RSP, Haberberger J, Severson E, Duncan DL, Hemmerich A, Edgerly C, Ferguson NL, Williams E, Elvin J, Vergilio JA, et al. A pan-cancer analysis of PD-L1 immunohistochemistry and gene amplification, tumor mutation burden and microsatellite instability in 48,782 cases. Mod Pathol. 2021;34(2):252–63.
Schildhaus HU. Predictive value of PD-L1 diagnostics. Pathologe. 2018;39(6):498–519
Hendry S, Byrne DJ, Wright GM, Young RJ, Sturrock S, Cooper WA, et al. Comparison of Four PD-L1 Immunohistochemical Assays in Lung Cancer. J Thorac Oncol. 2018 Mar 1;13(3):367–76.
Tsimafeyeu I, Imyanitov E, Zavalishina L, Raskin G, Povilaitite P, Savelov N, Kharitonova E, Rumyantsev A, Pugach I, Andreeva Y, et al. Agreement between PDL1 immunohistochemistry assays and polymerase chain reaction in non-small cell lung cancer: CLOVER comparison study. Sci Rep. 2020;10(1).
Guo H, Ding Q, Gong Y, Gilcrease MZ, Zhao M, Zhao J, Sui D, Wu Y, Chen H, Liu H, et al. Comparison of three scoring methods using the FDA-approved 22C3 immunohistochemistry assay to evaluate PD-L1 expression in breast cancer and their association with clinicopathologic factors. Breast Cancer Res. 2020;22(1):1–18.
Reviewer 3 Report
Reviewers Comments:
The authors present an interesting study in the review article entitled “Analysis of PD-L1 and its predictive and prognostic value in breast cancer.”. I think it would be worthwhile making major adjustments for acceptance in the “Cancers” Journal.
Major Comment:
1) Review article need to be strictly modified in a Review format.
Eg: The introduction, Objective, methodology, and outcome need to be concisely summarized in the abstract.
Subsections need to be apt for the main sections.
Eg: 5.1. Proposal PD-L1 analysis in breast cancer: The point mentioned in this section is not clear and needs to edit the language. This section can be a continuation of the conclusion.
2) Result section can contribute 90 % of the article with relevant subsections. Please follow the regular review format to modify the article.
3) The review article does not explain the need for PD-L1 therapy in comparison to other available therapies in reference to heterogenous and different subtypes of Breast cancer? Please add the relevant research studies and also the disadvantage of PD-L1 therapy (toxicities) in preclinical and clinical models?
4) Extensive editing is required for language and typographical errors?
Author Response
REVIEWER 3
- First of all, we would like to thank you for your corrections. Following your recommendations, we have now prepared a thoroughly revised version of the manuscript, in which we have addressed all the concerns.
We sincerely appreciate the comments and believe these revisions will improve the quality of our manuscript.
The authors present an interesting study in the review article entitled “Analysis of PD-L1 and its predictive and prognostic value in breast cancer.”. I think it would be worthwhile making major adjustments for acceptance in the “Cancers” Journal.
Major Comment:
1) Review article need to be strictly modified in a Review format.
Eg: The introduction, Objective, methodology, and outcome need to be concisely summarized in the abstract.
Subsections need to be apt for the main sections.
Eg: 5.1. Proposal PD-L1 analysis in breast cancer: The point mentioned in this section is not clear and needs to edit the language. This section can be a continuation of the conclusion.
-
Following the reviewer instructions, we have created a simple summary and modified the abstract, (lines 14-47). Moreover, we have removed section 5.1, including it within the conclusions (lines 459-476).
2) Result section can contribute 90 % of the article with relevant subsections. Please follow the regular review format to modify the article.
-
To follow the regular review format, section 4 (line 92) is titled Results and discussion. We have modified the sections to make them more understandable for the readers (pages 3-10).
3) The review article does not explain the need for PD-L1 therapy in comparison to other available therapies in reference to heterogeneous and different subtypes of Breast cancer? Please add the relevant research studies and also the disadvantage of PD-L1 therapy (toxicities) in preclinical and clinical models?
-
Authors have included some extra information about the heterogeneity in TNBC and different treatment options.
-
Lines 434-446: TNBC phenotype is very heterogeneous and has different biological behaviors. In the last few years, a great effort has been made in order to better characterize the diverse types of TNBC based on their somatic mutational profile, to find possible therapeutic targets [63,64]. The immunomodulatory TNBC subtype shows an increase in the expression of genes involved in immune signaling pathways, including PD-1/PD-L1 pathway [65], being immunotherapy a good treatment option in this subgroup.
At the present time, for PD-L1 positive advanced TNBC patients, the preferred option is ChT and immunotherapy combination. In case of PD-L1 positivity assessed by VENTANA SP142 the new standard is nab-paclitaxel plus atezolizumab. In case of PD-L1 CPS≥10 measured by DAKO 22C3, the treatment consists of ChT plus pembrolizumab. Other first-line treatment options such as olaparib, talazoparib or ChT with carboplatin for BRCA mutated patients, and ChT including or not bevacizumab for PD-L1 negative patients are available [66].
-
The references are:
Lehmann BD, Pietenpol JA. Identification and use of biomarkers in treatment strategies for triple-negative breast cancer subtypes. Journal of Pathology. 2014. Jan;232(2):142-50.
Lehmann BD, Jovanović B, Chen X, Estrada M V., Johnson KN, Shyr Y, Moses HL, Sanders ME, Pietenpol JA. Refinement of triple-negative breast cancer molecular subtypes: Implications for neoadjuvant chemotherapy selection. PLoS One. 2016 Jun 16;11(6):e0157368.
Yam C, Mani SA, Moulder SL. Targeting the Molecular Subtypes of Triple Negative Breast Cancer: Understanding the Diversity to Progress the Field. Oncologist. 2017 Sep;22(9):1086-1093.
Gennari A, André F, Barrios CH, Cortés J, De Azambuja E, Demichele A, Dent R, Fenlon D, Gligorov J, Hurvitz SA. ESMO Clinical Practice Guideline for the diagnosis, staging and treatment of patients with metastatic breast cancer 5 behalf of the ESMO Guidelines Committee. Ann Oncol. 2021;32:1475–95.
-
On the other hand, we would like to remark that the aim of this review does not include the toxicities or side effects of the immunotherapy.
4) Extensive editing is required for language and typographical errors?
The text has been proofread by native English speaker. Moreover, we have corrected some typographical errors in the text, and redacted the bibliography according to the journal system.
Round 2
Reviewer 1 Report
The authors have addressed all the concerns raised by this reviewer. The manuscript may be accepted for publication.
Author Response
We sincerely appreciate the comments and believe these revisions will improve the quality of our manuscript.